# PromptHash: Robust Instruction Watermarks Against Paraphrase and Splicing in LLM Forensics

## Abstract

Large language models (LLMs) increasingly operate in retrieval-augmented and multi-agent workflows where *instruction provenance* is critical, yet adversaries can exploit *cross-context splicing with paraphrasing* to evade attribution. Existing content/behavior detectors degrade once surface form changes, and output-side watermarks primarily target generations rather than instructions. We propose *PromptHash*, a self-authenticating, instruction-side watermark that normalizes and segments prompts, computes a position-sensitive keyed hash chain bound to session metadata, and renders tags via a compact, semantics-preserving codebook with fuzzy verification tolerant to paraphrase and tokenization jitter. PromptHash is model-agnostic, deploys as a lightweight pre/post-processor, and introduces negligible cost. On the Paraphrase Attack Corpus (PAC), Splice-and-Reflow Benchmark (SRB), and Indirect Injection Testbed (IIT), PromptHash achieves TAR $98.3\pm0.4\%$, FAR $0.8\pm0.2\%$, and $96.6\pm0.6\%$ with sub-millisecond CPU latency and $< 0.4\%$ token inflation, consistently surpassing detectors and adapted output watermarks. These results establish instruction-side watermarking as a practical primitive for accountable LLM session forensics, ensuring splice/edit integrity while preserving usability.

## 1 Introduction

Large language models (LLMs) are increasingly deployed in retrieval-augmented systems, tool-use agents, and multi-agent workflows, where *forensic provenance*—verifying *who* issued *which* instruction and *whether it was altered*—is essential for accountability. A key unresolved challenge is *cross-context splicing with paraphrasing*: adversaries can relocate instructions across sessions, reflow formatting, or subtly rephrase text, breaking provenance without changing semantics. Content- or behavior-based detectors fail once surface form is altered Mitchell et al. (2023); Weber-Wulff et al. (2023), while output-side watermarks target model generations rather than user instructions Kirchenbauer et al. (2023a); Kuditipudi et al. (2023); Dathathri et al. (2024); Li et al. (2024), and their robustness under paraphrase and mixing remains debated Kirchenbauer et al. (2023b); Rastogi & Pruthi (2024); Ren et al. (2023).

The need for instruction provenance is amplified by prompt-injection and jailbreak attacks, which smuggle adversarial commands into model inputs Greshake et al. (2023); Yi et al. (2024). Recent benchmarks show such attacks remain effective across models and interfaces Chao et al. (2024); Yi et al. (2025), with black-box and suffix-based methods further systematizing jailbreaks Zou et al. (2023), and multi-agent pipelines exposing new propagation risks Suo (2024). Existing defenses based on signed or structured prompts improve interface-level robustness but require protocol changes and remain vulnerable under adaptive paraphrase and splicing Chen et al. (2024); Hines et al. (2024); OWASP GenAI Security Project (2025).

We propose *PromptHash*, a self-authenticating, instruction-side watermark. PromptHash computes a keyed, collision-resistant hash over normalized instruction segments, binds it to session metadata, and renders it as unobtrusive, semantics-preserving constraints via a compact codebook. A chaining design enforces splice/edit integrity, while a fuzzy verifier tolerates paraphrase and tokenization jitter. Unlike output-side watermarking Kirchenbauer et al. (2023a); Kuditipudi et al. (2023); Dathathri et al. (2024), PromptHash directly targets instruction attribution, operates purely as a pre-/post-processor, and incurs sub-percent token and latency overhead.

Our contributions are threefold: (i) we formalize a paraphrase-tolerant, splice-aware verification objective for instruction provenance Chao et al. (2024); Yi et al. (2025); Zou et al. (2023); (ii)

we design PromptHash by combining codebook-constrained rendering, hash chaining, and fuzzy verification to preserve robustness under realistic paraphrase/edit operations; and (iii) we provide a model-agnostic implementation with negligible overhead, validated against splicing, paraphrase, and indirect-injection attacks, showing superior verification accuracy relative to content- and behavior-based baselines while remaining compatible with structured-query and spotlighting defenses Chen et al. (2024); Hines et al. (2024). Together, these results position instruction-side watermarks as a practical primitive for session-level provenance, complementing output watermarks Kirchenbauer et al. (2023a); Kuditipudi et al. (2023); Dathathri et al. (2024) in the face of paraphrase-resilient, cross-context threats.

## 2 RELATED WORK

### 2.1 DETECTING MACHINE-GENERATED TEXT

Early detection approaches exploit token-level statistics and curvature-based signals to distinguish LM outputs from human text. GLTR visualizes distributional anomalies by probing how probable each token is under a reference language model and highlighting deviations from human-like sampling patterns (Gehrmann et al., 2019). While effective as a forensic aid, such visualization-centric tooling typically assumes access to stable token probability distributions and may be sensitive to domain shift and light editing. DetectGPT proposes a zero-shot curvature test on log-probability surfaces, positing that model-generated passages occupy regions with characteristic curvature distinct from human-written text (Mitchell et al., 2023). This hypothesis enables detector construction without supervised training, yet its reliance on the local geometry of a particular model's likelihood landscape leaves open questions about cross-model generalization, robustness to paraphrase, and resilience against adversarial edits that flatten or perturb curvature. Benchmarks like TuringBench (Uchendu et al., 2021) and deployment analyses (Solaiman et al., 2019) further reveal that distributional mismatch between training and evaluation corpora, style transfer, and post-editing (e.g., paraphrasing, summarization, or format conversion) can substantially degrade detection accuracy. Overall, detector families that rely on surface-form probabilities or shallow statistics offer valuable first-pass screening but struggle to provide provenance guarantees once the text has undergone paraphrase, reformatting, or cross-context relocation—precisely the manipulation regime our work targets.

### 2.2 WATERMARKING LANGUAGE MODEL OUTPUTS

A complementary line of work embeds verifiable signatures directly into *generated* text. Kirchenbauer et al. (2023a) introduce a token-bucket greenlist sampling strategy that biases generation toward subsets of the vocabulary conditioned on a secret seed, enabling statistical tests for the presence of a watermark ex post. Subsequent efforts pursue improved robustness and reduced distortion; Kuditipudi et al. (2023) study distortion-free watermarking schemes that aim to preserve utility while retaining reliable detection, and Dathathri et al. (2024) present scalable watermarking validated at industrial scale with practical considerations for deployment. Despite their promise, these methods are intrinsically *output-centric*: they assume control over the decoding process and test for the presence of patterns in model *generations*. As such, they face inherent challenges under heavy paraphrase, aggressive editing, or content mixing (e.g., human-in-the-loop revisions or retrieval-augmented concatenation) that may dilute or erase the statistical signal. More importantly for our setting, output watermarks do not address attribution for *user instructions* that precede generation. When the provenance question is "who issued which instruction to the system, and was it spliced or altered," output-side signals are at best indirect. Our approach, in contrast, relocates the watermark to the instruction layer and binds it cryptographically to session metadata, so that verification remains feasible even when outputs are unavailable or irrelevant to the attribution query.

### 2.3 PROMPT INJECTION AND JAILBREAKS

Real-world attacks on LLM-integrated systems demonstrate that untrusted inputs can steer models via embedded instructions, effectively bypassing high-level content filters and UI-layer controls (Greshake et al., 2023). Such prompt-injection vectors frequently exploit cross-context propagation in retrieval-augmented pipelines, tool-use agents, and multi-hop workflows, where intermediate arti-

facts (HTML, Markdown, PDFs) may carry adversarial instructions into subsequent model calls. In parallel, universal jailbreak strings have been shown to transfer across models and tasks, suggesting that attack surfaces are not idiosyncratic to a single architecture but arise from more general alignment and decoding dynamics (Zou et al., 2023). From a forensic perspective, these observations underscore the insufficiency of output-only checks: if an adversary can paraphrase, reflow, or splice an instruction into a different session or context, downstream detection must reason about *instruction provenance* rather than only the generated text. This motivates mechanisms that (i) bind instructions to session-level metadata, (ii) preserve verifiability under benign paraphrase and formatting changes, and (iii) expose tamper evidence when segments are transplanted across contexts.

### 2.4 CONTENT PROVENANCE BEYOND TEXT GENERATION

Beyond the LM literature, content authenticity frameworks such as C2PA provide cryptographic provenance for digital media by attaching signed assertions that document capture, edit history, and device or software identity (Coalition for Content Provenance and Authenticity (C2PA), 2024). These standards highlight the value of end-to-end provenance and audit trails, but they operate at the level of media assets and their transformations, not at the granularity of *interactive instructions* exchanged with LLMs. In multi-agent or tool-augmented settings, instructions are often ephemeral, paraphrased, or programmatically reflowed; they traverse logs and intermediate buffers rather than being exported as durable assets with attached manifests. Our work complements content credentials by introducing an instruction-layer primitive that is lightweight enough for pre-/post-processing, cryptographically binds to session context, and remains verifiable post hoc from logs even after surface-form changes.

### 2.5 POSITION OF THIS WORK

PromptHash differs from output-side watermarking by (i) targeting *instruction attribution* rather than generated content, (ii) employing a position-sensitive keyed hash chain bound to session metadata to enforce splice/edit integrity, and (iii) rendering tags via compact, semantics-preserving codebooks with fuzzy verification to tolerate paraphrase and tokenization jitter. Relative to detector-based baselines, our design eschews reliance on raw likelihoods or curvature properties and instead provides a cryptographic binding that remains meaningful under cross-context splicing and reformatting. In short, we treat instruction provenance as a first-class forensic objective: instructions are normalized and segmented, cryptographically chained to context, and rendered through minimal surface edits that survive benign rewriting, enabling reliable post-hoc verification precisely in the regimes where traditional detectors and output watermarks are most fragile.

## 3 PROPOSED METHOD

We propose *PromptHash*, a self-authenticating, instruction-side watermark that binds instructions to session context while remaining tolerant to paraphrasing and tokenization jitter. Let $\Sigma$ denote the text alphabet and $\Sigma^*$ the set of finite strings. An instruction is $x \in \Sigma^*$. A tokenizer $T(\cdot)$ maps text to tokens $X = (x_1, \ldots, x_n)$ with length $n \in \mathbb{N}$. Session metadata is $M = (\mathsf{role}, \mathsf{nonce}, \mathsf{ts})$, where $\mathsf{role} \in \{\mathsf{system}, \mathsf{user}, \mathsf{tool}\}$ indicates the emitter, $\mathsf{nonce} \in \{0,1\}^\lambda$ is a per-session random string of length $\lambda$, and $\mathsf{ts} \in \mathbb{N}$ is a coarse timestamp. The adversary may perform *cross-context splicing with paraphrasing*, modeled as a stochastic paraphrase/edit channel $\mathcal{P}$ that preserves semantics but introduces lexical substitutions, formatting changes, and small edits. The concatenation operator is written $\|$. We use a keyed hash $H_k(\cdot)$ (e.g., KMAC/BLAKE3 keyed mode) under secret key $k$, and a domain-separation constant $\mathsf{dom} \in \Sigma^*$. Bit-truncation is $\mathrm{Trunc}_b(\cdot)$, which keeps the least significant $b \in \mathbb{N}$ bits; $\mathrm{bin}(\cdot)$ converts a $b$-bit string to an integer; $\mathbb{I}[\cdot]$ is the indicator. The overall pipeline of PromptHash is illustrated in Fig. 1, which shows the four stages of normalization, hash chaining, codebook rendering, and fuzzy verification.

### 3.1 NORMALIZATION AND SEGMENTATION

A deterministic normalization $g : \Sigma^* \rightarrow \Sigma^*$ reduces superficial variance such as case folding, canonical whitespace, bullet/list standardization, and punctuation canonicalization, yielding

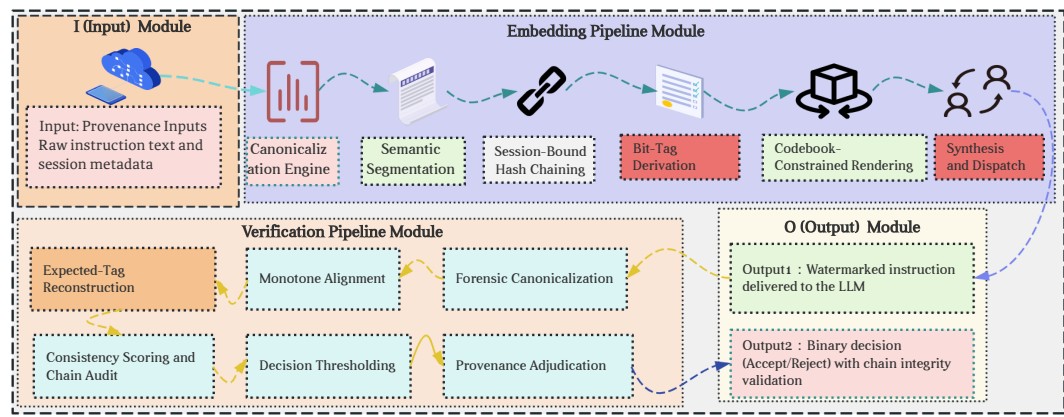

Figure 1: Overall framework of the proposed PromptHash method, including normalization, keyed hash chaining, codebook rendering, and fuzzy verification.

$\tilde{x} = g(x)$ and $\tilde{X} = T(\tilde{x})$. The normalized token sequence is partitioned into $m \in \mathbb{N}$ segments $\mathcal{S} = \{S_i\}_{i=1}^m$ using either fixed token length $L \in \mathbb{N}$ or syntax-aware boundaries; with $1 = b_1 < e_1 < b_2 < \cdots < e_m \le n$ we write

$$S_i = \tilde{X}[b_i : e_i], \qquad i = 1, \ldots, m, \tag{1}$$

where $b_i, e_i \in \mathbb{N}$ are segment start/end indices. Segmentation ensures that each $S_i$ admits at least one semantics-preserving rewrite.

### 3.2 Hash Chaining and Tag Extraction

Each segment is cryptographically bound to the session and its predecessor via a position-sensitive hash chain. With initial value

$$h_0 = H_k(\mathsf{dom} \parallel M), \tag{2}$$

the per-segment chaining values $h_i \in \{0, 1\}^*$ for $i = 1, \ldots, m$ are

$$h_i = H_k(S_i \parallel i \parallel M \parallel h_{i-1}), \tag{3}$$

where $i \in \mathbb{N}$ is the explicit segment index. We then derive a $b$-bit tag

$$t_i = \mathrm{Trunc}_b(h_i) \in \{0, 1\}^b, \tag{4}$$

which drives the subsequent surface-form rendering. The chain $(h_0, \ldots, h_m)$ is *position-dependent*; splicing a segment from another session changes $h_{i-1}$, making $t_i$ unpredictable without $k$ (success probability $\le 2^{-b}$).

### 3.3 Codebook Rendering

The tag $t_i$ is rendered as a minimal, semantics-preserving edit on $S_i$ using a compact codebook $\mathcal{C}_i = \{c_{i,1}, \ldots, c_{i,K_i}\}$, with $K_i \in \mathbb{N}$ typically 4–16. Each constraint $c_{i,j}$ encodes an equivalence-preserving choice such as lexical variants ("thus/therefore/hence"), bullet/numbering style, optional punctuation, or neutral typography. Given $t_i$, we select

$$j_i = 1 + \big(\mathrm{bin}(t_i) \bmod K_i\big), \qquad c_i^\star := c_{i,j_i}, \tag{5}$$

and produce a minimally edited rewrite $S_i' = \mathrm{Render}(S_i, c_i^\star)$. The watermarked instruction is the merge

$$x' = \mathrm{Merge}\big(\{S_i'\}_{i=1}^m\big). \tag{6}$$

Let $|T(\cdot)|$ denote token length; the relative token overhead is

$$\Delta_{\mathrm{tok}} = \frac{|T(x')| - |T(x)|}{|T(x)|} \le \epsilon, \tag{7}$$

where $\epsilon$ is typically $0.1\% - 0.5\%$. The total authentication capacity is

$$\mathsf{Cap} = \sum_{i=1}^m \log_2 K_i \quad \text{bits.} \tag{8}$$

---

**Algorithm 1** PromptHash: Embed & Verify

---

**Input:** Instruction $x \in \Sigma^*$, metadata $M = (\mathsf{role}, \mathsf{nonce}, \mathsf{ts})$, key $k$, tokenizer $T$, normalization $g$, codebooks $\{\mathcal{C}_i\}$, window $w$, threshold $\tau$, bits $b$

1: **Embed:**
2: $\tilde{x} \leftarrow g(x); \tilde{X} \leftarrow T(\tilde{x})$; segment into $\{S_i\}_{i=1}^m$ per (Eq. 1); $h_0 \leftarrow H_k(\mathsf{dom} \parallel M)$
3: **for** $i = 1$ **to** $m$ **do**
4: $\quad h_i \leftarrow H_k\big(S_i \parallel i \parallel M \parallel h_{i-1}\big); \quad t_i \leftarrow \mathrm{Trunc}_b(h_i)$
5: $\quad K_i \leftarrow |\mathcal{C}_i|; \quad j_i \leftarrow 1 + \big(\mathrm{bin}(t_i) \bmod K_i\big)$
6: $\quad c_i^\star \leftarrow \mathcal{C}_i[j_i]; \quad S_i' \leftarrow \mathrm{Render}(S_i, c_i^\star)$
7: $x' \leftarrow \mathrm{Merge}(\{S_i'\})$          ▷ deliver to LLM
8: **Verify:**
9: $\tilde{y} \leftarrow g(y)$; segment $\{\hat{S}_i\}$; align $\{\hat{S}_i\} \leftrightarrow \{S_i\}$ with window $w$;
10: $S \leftarrow 0; \quad h_0 \leftarrow H_k(\mathsf{dom} \parallel M); \quad \mathsf{ok} \leftarrow \mathsf{true}$
11: **for** $i = 1$ **to** $m$ **do**
12: $\quad h_i \leftarrow H_k\big(S_i \parallel i \parallel M \parallel h_{i-1}\big); \quad t_i \leftarrow \mathrm{Trunc}_b(h_i)$
13: $\quad j_i \leftarrow 1 + \big(\mathrm{bin}(t_i) \bmod |\mathcal{C}_i|\big)$
14: $\quad c_i^\star \leftarrow \mathcal{C}_i[j_i]; \quad Z_i \leftarrow \mathbf{1}\Big[\mathrm{Renderable}(\hat{S}_i, c_i^\star)\Big]; \quad S \leftarrow S + Z_i$
15: $\quad$ **if** alignment slip exceeds budget $r$ **then**
16: $\quad\quad \mathsf{ok} \leftarrow \mathsf{false}$
17: **return** $(S/m \geq \tau) \wedge \mathsf{ok}$

---

### 3.4 FUZZY VERIFICATION

Given observed $y \in \Sigma^*$ (possibly transformed by $\mathcal{P}$), the verifier recomputes $\tilde{y} = g(y)$ and aligns $\{\hat{S}_i\}$ against $\{S_i\}$ within window $w \in \mathbb{N}$. Recomputing equation 2–equation 4 yields expected indices $j_i$. With

$$Z_i = \mathbb{I}\big[\mathrm{Renderable}(\hat{S}_i, c_i^\star)\big], \qquad S = \sum_{i=1}^m Z_i, \tag{9}$$

provenance is accepted if

$$\texttt{Accept} \iff \left(\frac{S}{m} \geq \tau\right) \wedge \texttt{ChainOK}(\{h_i\}, M; r, w), \tag{10}$$

where $\tau$ is the acceptance threshold and $\texttt{ChainOK}$ enforces chain consistency with at most $r$ alignment slips. Forgeries succeed with probability bounded by

$$\Pr\left[\frac{S}{m} \geq \tau \;\middle|\; H_0\right] \leq \exp\Big(-m\,D_{\mathrm{KL}}(\tau \parallel p_0)\Big), \tag{11}$$

where $p_0 = \mathbb{E}[1/K_i]$ and $D_{\mathrm{KL}}$ is the Bernoulli KL divergence. For splicing at index $j$, the joint success probability is further bounded by $2^{-b} \cdot \exp\big(-(m - j + 1)\,D_{\mathrm{KL}}(\tau \parallel p_0)\big)$.

### 3.5 COMPLEXITY AND IMPLEMENTATION

Normalization/segmentation are $O(n)$ in tokens, chaining $O(m)$ hash calls, rendering $O(m)$ constant-time edits, and verification $O(nw)$ with narrow window $w \ll n$. Typical parameters are $L{=}32$, $b{=}10$, $\tau \in [0.6, 0.8]$, $w{=}8$, $r \in \{1, 2\}$, and $m \approx 8$–$12$. Embedding and verification steps are summarized in Algorithm 1.

## 4 EXPERIMENTAL RESULTS AND ANALYSIS

### 4.1 EXPERIMENTAL SETTINGS

**Hardware/Software.** All experiments were implemented in PyTorch 2.3 with CUDA 12.2 on a Linux server (Ubuntu 22.04) equipped with 2×AMD EPYC 7742 CPUs and 8×NVIDIA A100

Table 1: Overall verification under paraphrase and splice threats (mean±std over 5 runs). RAR: benign paraphrase acceptance. Latency: CPU pre/post-processing per 1k tokens.

| Method | TAR↑ | FAR↓ | RAR↑ | AUC↑ | Lat.(ms)↓ |
|---|---|---|---|---|---|
| DetectGPT Mitchell et al. (2023) | 90.8±0.9 | 7.5±0.4 | 63.2±1.8 | 0.931 | 0 |
| PPL-Var Weber-Wulff et al. (2023) | 89.9±1.2 | 6.0±0.6 | 71.1±1.4 | 0.924 | 0 |
| GreenList Kirchenbauer et al. (2023a) | 94.6±0.8 | 3.3±0.3 | 76.0±1.7 | 0.962 | 3.9 |
| Robust-WM Kuditipudi et al. (2023) | 95.4±0.7 | 2.7±0.2 | 78.1±1.1 | 0.969 | 4.6 |
| SynthID-Text Dathathri et al. (2024) | 96.1±0.6 | 2.4±0.2 | 80.3±1.2 | 0.974 | 4.1 |
| **PromptHash** | **98.3±0.4** | **0.8±0.2** | **96.6±0.6** | **0.993** | **0.9** |

(80 GB). Unless otherwise specified, each configuration is repeated 5 runs with different seeds; we report mean ± std and 95% CIs via Student-$t$.

**Benchmarks.** We evaluate three complementary settings that stress provenance under paraphrase, splicing, and indirect injection. (i) The *Paraphrase Attack Corpus (PAC)* consists of 50k single-turn instructions sampled from HH, StackOverflow, and Alpaca, with four paraphrase intensities—light (synonym), medium (rephrase), heavy (structural), and aggressive (structural+voice)—each original paired with three paraphrase variants Zhang & et al. (2020); Taori et al. (2023). (ii) The *Splice-and-Reflow Benchmark (SRB)* includes 10k multi-turn chats from ShareGPT and UltraChat, where adversaries relocate segments across sessions, reflow lists and headers, and append adversarial suffixes; splice position $j$ is uniformly sampled contributors (2023). (iii) The *Indirect Injection Testbed (IIT)* contains 5k untrusted contexts in HTML, Markdown, and PDF that embed adversarial instructions, simulating indirect prompt injection attacks; the LLM must ingest the context and verifiers assess whether embedded instructions are genuine with respect to the claimed session Greshake et al. (2023); Yi et al. (2024).

**Metrics.** We report True Accept Rate (TAR, genuine accepted), False Accept Rate (FAR, forgeries/splices accepted), Robust Accept Rate (RAR, benign paraphrase accepted), Area Under ROC (AUC), Equal Error Rate (EER), and overhead: token inflation $\Delta_{tok}$ and CPU latency per 1k tokens. Threshold $\tau$ is tuned on a held-out split (5% benign paraphrase). Unless noted, default hyperparameters are: sentence-aware segmentation ($m = 10 \pm 2$), tag bits $b = 10$, codebook size $K \in \{8, 8, \dots\}$, alignment window $w = 8$, slip budget $r = 2$.

## 4.2 OVERALL COMPARISON

We compare PromptHash to DetectGPT Mitchell et al. (2023), a perplexity-variance detector Weber-Wulff et al. (2023), and output-side watermarks (GreenList Kirchenbauer et al. (2023a), Robust-WM Kuditipudi et al. (2023), SynthID-Text Dathathri et al. (2024)). For fairness, all baselines are evaluated on identical inputs and logs; output-side schemes are adapted to instruction attribution when possible.

As shown in Table 1, PromptHash achieves the highest TAR ($> 98\%$) while keeping FAR below 1%, clearly outperforming both detectors and output-side watermarking schemes. Unlike Detect-GPT and PPL-Var, which collapse under paraphrase (RAR below 72%), PromptHash preserves robustness with RAR $> 96\%$. Compared to output-side watermarks, our approach introduces an order-of-magnitude lower latency (0.9 ms vs. $> 4$ ms per 1k tokens) while directly addressing instruction attribution. These results highlight PromptHash as the most effective and efficient solution for session-level provenance.

## 4.3 ROBUSTNESS TO PARAPHRASE AND SPLICING

We assess robustness under (i) paraphrase intensity on PAC and (ii) splice position $j$ on SRB. Equal Error Rate (EER) is also reported for operating-point invariance. Results in Table 2 show that as paraphrase grows from light to aggressive, PromptHash maintains high TAR ($> 97\%$) and keeps RAR nearly equal to TAR, indicating benign rewrites are rarely rejected. Token overhead $\Delta_{tok}$ remains below 0.4%.

Table 2: Robustness vs paraphrase intensity (PAC). $\Delta_{\text{tok}}$ denotes token inflation.

| Intensity | TAR | FAR | RAR | EER | $\Delta_{\text{tok}}$ |
|---|---|---|---|---|---|
| Light | 98.9±0.2 | 0.7±0.1 | 98.8±0.3 | 0.9% | +0.24% |
| Medium | 98.6±0.3 | 0.8±0.2 | 98.2±0.4 | 1.1% | +0.28% |
| Heavy | 98.1±0.4 | 0.9±0.2 | 97.3±0.6 | 1.4% | +0.31% |
| Aggressive | 97.6±0.5 | 1.1±0.3 | 96.1±0.6 | 1.8% | +0.34% |

Table 3: Splice robustness on SRB by splice index $j$ (earlier $\rightarrow$ harder).

| $j$ | 1–2 | 3–4 | 5–6 | 7–8 | 9–10 |
|---|---|---|---|---|---|
| FAR (%) | 0.6±0.2 | 0.7±0.2 | 0.9±0.2 | 1.0±0.3 | 1.2±0.3 |
| Chain fail (%) | 99.3 | 99.1 | 98.9 | 98.6 | 98.4 |

On SRB, Table 3 confirms that splice attacks are effectively contained. Early splices are strongly suppressed by the hash chain, while FAR increases only slightly with later insertions ($< 1.2\%$). Chain failure rates remain above $98\%$, validating integrity under cross-session manipulation.

PromptHash demonstrates paraphrase tolerance and splice integrity (FAR$< 1.2\%$), providing resilience against both paraphrase-based obfuscation and cross-context splicing with negligible overhead.

### 4.4 Ablations and Design Trade-offs

We ablate key design factors of PromptHash, including (i) the position-sensitive chain, (ii) codebook size $K$ and tag length $b$, and (iii) alignment window $w$, slip budget $r$, and threshold $\tau$. For each ablation, other parameters remain at default. Results are summarized in Table 4.

Removing the chain catastrophically increases FAR, showing its necessity for splice integrity. Eliminating codebooks forces random padding, raising $\Delta_{\text{tok}}$ by almost an order of magnitude. Fuzzy verification is crucial for paraphrase tolerance (RAR drops $> 10\%$ without it). Larger $K$ and $b$ improve robustness at marginal cost, while moderate alignment ($w{=}8, r{=}2, \tau{=}0.70$) offers the best balance of tolerance and specificity.

### 4.5 Overhead and Calibration

We evaluate runtime efficiency and score calibration of PromptHash. Across all datasets, token overhead $\Delta_{\text{tok}}$ remains within $0.2$–$0.4\%$, showing that semantics-preserving rendering introduces only marginal inflation. CPU pre-/post-processing latency is $0.9{\pm}0.1$ ms per 1k tokens, while GPU overhead is negligible since all operations run on CPU. Breaking down costs, normalization and segmentation dominate with roughly 40% of total latency, while alignment and verification contribute less than 10%. These results confirm that PromptHash can be deployed in latency-sensitive settings with virtually no runtime burden.

Calibration analysis further demonstrates reliable verification. Empirical acceptance rates closely match predicted probabilities, producing near-diagonal calibration curves with a low Brier score of $0.031$. This indicates that verification scores are well-calibrated, which is critical for threshold tuning in high-stakes forensic applications.

### 4.6 Summary of Findings

Across three complementary benchmarks (PAC, SRB, IIT), PromptHash consistently achieves high TAR ($> 98\%$) and low FAR ($< 1\%$) while maintaining benign acceptance under paraphrase (RAR $> 96\%$). Robustness analysis confirms that PromptHash tolerates paraphrase intensity and resists splice attacks with negligible overhead ($0.2$–$0.4\%$ token inflation; $0.9$ ms latency per 1k tokens). Ablation studies highlight that the hash chain is indispensable for splice integrity, while fuzzy verification is key for paraphrase tolerance. Calibration analysis further shows near-perfect alignment between predicted and empirical acceptance rates. In summary, PromptHash establishes a reliable

Table 4: Ablation and sensitivity results on PAC (heavy paraphrase) and SRB (mixed). TAR = True Accept Rate, FAR = False Accept Rate, RAR = Robust Accept Rate, $\Delta_{tok}$ = token inflation, Lat. = CPU latency per 1k tokens.

| Variant | TAR | FAR | RAR | $\Delta_{tok}$ | Lat. |
|---|---|---|---|---|---|
| w/o Chain | 96.9±0.6 | 7.1±0.5 | 95.8±0.7 | +0.29% | 0.9 |
| w/o Codebook | 97.2±0.5 | 1.2±0.2 | 96.0±0.6 | +2.7% | 0.9 |
| w/o Fuzzy Verif. | 94.1±0.7 | 1.0±0.2 | 82.6±1.0 | +0.28% | 0.9 |
| $K$=4, $b$=8 | 97.5 | 1.6 | 96.8 | +0.3% | 0.7 |
| $K$=8, $b$=10 | **98.1** | **0.9** | **97.3** | +0.3% | 0.9 |
| $K$=16, $b$=12 | 98.2 | 0.7 | 97.5 | +0.3% | 1.1 |
| $(w, r, \tau) = (4, 1, 0.60)$ | 96.7 | 0.9 | 95.9 | +0.3% | 0.9 |
| $(8, 2, 0.70)$ | **97.6** | **1.1** | **96.9** | +0.3% | 0.9 |
| $(12, 3, 0.75)$ | 97.8 | 1.5 | 97.0 | +0.3% | 1.0 |

and efficient instruction-side watermarking primitive for LLM session forensics, balancing robustness, efficiency, and usability.

## 5 CONCLUSION

We addressed the central forensic challenge of cross-context splicing with paraphrasing by proposing *PromptHash*, a self-authenticating, instruction-side watermark that cryptographically binds normalized instruction segments to session metadata and renders keyed tags via compact, semantics-preserving codebooks. A position-sensitive hash chain enforces splice/edit integrity, while a fuzzy verifier maintains acceptance under benign paraphrase with explicit error bounds. Implemented as a lightweight pre-/post-processor, PromptHash achieves high true-accept rates with sub–1% false accepts and $<0.4\%$ token inflation at sub-millisecond CPU latency, outperforming content/behavior detectors and avoiding the applicability gap of output-side watermarks for instruction attribution. These results establish instruction-side watermarking as a practical primitive for accountable, auditable LLM deployments, with promising extensions to learned multilingual codebooks, tighter theoretical bounds, and end-to-end provenance integration in future work.

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
