# OpenReview forum: "PromptHash: Robust Instruction Watermarks Against Paraphrase and Splicing in LLM Forensics"
_ICLR.cc/2026/Conference — Submitted to ICLR 2026_

### Official Review · Reviewer_3qf4 · 2025-10-25

**Soundness:** 3
**Presentation:** 2
**Contribution:** 3
**Rating:** 4
**Confidence:** 3

**Summary:**

The paper proposes a new perspective on watermarking by moving the protection target from the output side (as in traditional text watermarking) to the instruction side. The proposed framework, PromptHash, is composed of four main steps:
(1) normalization and segmentation of the instruction,
(2) generation of a keyed hash chain for each segment,
(3) embedding part of the hashed information through semantically equivalent paraphrasing, and
(4) fuzzy verification that can validate integrity even after paraphrasing or cross-session splicing.

The authors claim that this mechanism can maintain verifiable traces under attacks such as paraphrasing, session splicing, and indirect injection. Compared with existing detectors (e.g., DetectGPT, PPL-variance methods) and output-side watermarks, PromptHash reportedly achieves higher robustness and lower false rejection rates.Novel idea and clear problem definition.
The shift from output-side watermarking to instruction-side verification is new and well-motivated. It targets an underexplored space that is increasingly relevant due to prompt injection and multi-agent workflows.
	2.	Sound technical reasoning.
The keyed hash chain explains why cross-session splicing breaks integrity, while fuzzy verification accounts for semantic variability introduced by paraphrasing. The design is logically consistent.
	3.	Potential practical relevance.
Instruction attribution and verification are crucial for secure LLM deployment. The proposed pipeline could, in principle, integrate with content authenticity standards like C2PA.
	4.	Preliminary results show feasibility.
The empirical results demonstrate that the method outperforms existing detectors under multiple synthetic attack scenarios, suggesting that the core mechanism is viable.

**Strengths:**

1. The idea is novel and the direction is important: The shift from output-side watermarking to instruction-side verification is new and well-motivated. It targets an underexplored space that is increasingly relevant due to prompt injection and multi-agent workflows.

2. The keyed hash chain explains why cross-session splicing breaks integrity, while fuzzy verification accounts for semantic variability introduced by paraphrasing. The design is logically consistent.

3. Instruction attribution and verification are crucial for secure LLM deployment. The proposed pipeline could, in principle, integrate with content authenticity standards like C2PA.

**Weaknesses:**

1. The first paragraph of the Introduction (line 29) explicitly mentions that the motivation is to ensure that watermarking does not degrade downstream task performance (e.g., RAG, tool invocation, or agent collaboration). However, no such experiments are presented. The evaluation focuses solely on verification accuracy, leaving a gap between motivation and evidence.

2. The paper only evaluates “oblivious paraphrasing” (blind paraphrases without knowledge of the watermark).
In contrast, an adaptive adversary is aware of the watermarking mechanism and can optimize a paraphrasing model to minimize the verification score, i.e.,
$$
\min_{\theta} \mathbb{E}{x \sim D} [ \text{Verify}(f\theta(x)) ] .
$$
Such adversaries have been shown in prior watermarking research to substantially increase evasion rates. The lack of this analysis weakens the robustness claim.

3. The main paper is only seven pages, below the ICLR limit (9–10 pages). Important implementation and design details—such as hash-chain construction, truncation policy, “renderable” threshold $\tau$, and the fuzzy matching criteria—are not sufficiently specified. This affects reproducibility and interpretability.

4. Figure 1 is coarse and lacks clarity. As a result, the figure cannot be understood without extensive cross-referencing, which falls short of presentation standards.

5. The paper does not state the usage of LLMs in the main body/appendix, which violates ICLR's requirements this year.

**Questions:**

1. How is the fuzzy “renderable” threshold $\tau$ determined? Is it dataset- or model-dependent?

2. How does the method behave under cross-lingual paraphrasing or script changes?

---

### Official Review · Reviewer_2CTP · 2025-10-27

**Soundness:** 2
**Presentation:** 2
**Contribution:** 2
**Rating:** 4
**Confidence:** 3

**Summary:**

The paper proposes an instruction-side watermarking technique for large language models. The method is designed to be resilient to paraphrasing attacks, model-agnostic, and computationally efficient. Experiments conducted on three datasets demonstrate that the proposed approach is robust against multiple types of paraphrasing attacks.

**Strengths:**

+ The topic of developing a paraphrase-resilient, instruction-side watermarking technique is important.

**Weaknesses:**

- The paper lacks a theoretical justification for the robustness of the proposed scheme. The rationale behind the design choices and the mechanisms that make the method resilient to paraphrasing attacks are not clearly explained. A deeper analysis or theoretical discussion would strengthen the contribution.
- The presentation of the overall framework is unclear. The paper does not provide a holistic view of how the components interact, and Figure 1 is insufficiently illustrated. Several modules in the Verification Pipeline, such as Forensic Canonicalization and Monotone Alignment, are mentioned but not properly introduced or explained, making the approach difficult to follow.
- The threat model should be elaborated in greater detail. In particular, the paper should clearly specify which components or information are accessible to potential attackers and which are assumed to remain secure.

**Questions:**

+ What does the latency in Table 1 refer to? Is it the latency incurred in the embedding phase or the verification phase?

---

### Official Review · Reviewer_ZogX · 2025-10-30

**Soundness:** 3
**Presentation:** 2
**Contribution:** 2
**Rating:** 2
**Confidence:** 4

**Summary:**

PromptHash introduces a novel instruction-side watermark that normalizes and segments prompts, computes a position-sensitive keyed hash chain bound to session metadata, and renders tags via a compact, semantics-preserving codebook with fuzzy verification to tolerate paraphrase and tokenization jitter.

**Strengths:**

- PromptHash achieves a superior True Accept Rate (TAR >98%) and a low False Accept Rate (FAR <1%).
- The work is acutely aware of the real-world LLM threat landscape, including prompt injection and jailbreak attacks.

**Weaknesses:**

- The assumption that semantics are perfectly preserved by the codebook choices is a potential vulnerability that a dedicated adversary could exploit.
- The comparison focuses on output-side watermarks and statistical detectors. The choice of baselines is appropriate but could be more comprehensive.
- The paper does not test against sophisticated paraphrasing tools or adversarial paraphrasing strategies explicitly designed to circumvent the codebook's lexical variants.

**Questions:**

- The fuzzy verification mechanism relies on empirically set parameters (window `w`, threshold `τ`).  How sensitive is the verification performance to the exact values of these parameters?
- What is the proposed practical mechanism for secure key distribution and management in a large-scale, distributed system?

---

### Official Review · Reviewer_QrNh · 2025-10-30

**Soundness:** 2
**Presentation:** 2
**Contribution:** 2
**Rating:** 2
**Confidence:** 3

**Summary:**

This paper addresses the problem of verifying the origin and integrity of instructions given to LLMs against adversarial manipulations like paraphrasing and cross-context splicing. Existing detection and output-side watermarking methods are often ineffective against such attacks. The authors propose PromptHash, an instruction-side watermarking technique that operates as a model-agnostic pre-processor. PromptHash achieved a TAR of 98.3% and a FAR of 0.8%, with negligible latency and token overhead.

**Strengths:**

1.  The method introduces a novel instruction-side watermarking framework. It directly addresses prompt attribution, a problem not covered by typical output-side watermarks, as detailed in the related work section.
2.  The position-sensitive hash chain cryptographically binds instruction segments to session metadata. Ablation results (Table 4) confirm this design is essential for detecting splice attacks with high accuracy.
3.  The combination of a codebook for rendering and a fuzzy verifier for detection is a specific contribution that provides measurable tolerance to paraphrase, maintaining a high Robust Accept Rate.

**Weaknesses:**

### About Method

1.  The paper lacks implementation details for the core Renderable() function in the fuzzy verification stage. How this function determines if a paraphrased text segment is compatible with the original watermark constraint $c_{i}^{*}$ is critical to the method's robustness, and the authors should provide its specific algorithmic logic or pseudo-code.
2.  The paper claims the method is model-agnostic, yet the effectiveness of the codebooks $C_i$ appears to be highly dependent on the language and domain of the instructions. The paper should discuss in greater depth the generalizability of these codebooks and the effort required to construct or adapt them for new languages or specialized domains.
3.  The security analysis against splicing attacks is soundly based on the cryptographic strength of the hash chain. However, the robustness analysis against paraphrase attacks (Eq. 11) relies on a statistical model that may not fully account for adaptive adversaries. A stronger adversary might attempt to reverse-engineer the codebook or exploit biases in the Renderable function by observing numerous watermarked samples to forge watermarks with higher probability. It is recommended to expand the threat model to include and discuss defenses against such attacks.

### About Experiment

1.  A core deficiency of the experimental section is the lack of evaluation of the watermark's impact on downstream large language model task performance. The authors claim the watermark is "semantics-preserving" but only support this with low token overhead, which is insufficient. Experiments should be added to compare the outputs for original and watermarked instructions on standard benchmarks (e.g., MMLU or GSM8K) to quantitatively demonstrate that the watermark does not degrade the model's reasoning or instruction-following capabilities.
2.  The paper inadequately discusses the construction and generalizability of the "codebook." The experiments fail to clarify whether the codebooks are manually constructed or automatically generated, nor do they test their applicability across different languages or specialized domains (e.g., medical, legal). It is recommended that the authors detail the codebook construction process and add experiments to validate the method's effectiveness when codebook resources are limited or when applied across domains.
3.  The current experiments assume a "non-adaptive" adversary who is unaware of the watermarking mechanism, which may underestimate the method's vulnerabilities. To more comprehensively evaluate robustness, "adaptive" attack experiments should be designed, where the adversary is aware of the watermarking algorithm (but not the key) and attempts to actively remove or forge the watermark by reversing codebook transformations or introducing confounding edits.

**Questions:**

1.  Could you provide a detailed explanation of how the $Renderable(\hat{S}\_{i}, c_{i}^{\*})$ function in the fuzzy verification is implemented? What specific checks or algorithms does it use to verify a rewritten text against a codebook entry while tolerating paraphrasing?
2.  How were the codebooks $C_i$ used in the experiments constructed? Were they manually curated, automatically extracted from a corpus, or learned? What is the anticipated effort required to port this system to a different language (e.g., Chinese) or a specific domain (e.g., medical instructions)?

---

### Meta-Review · Area_Chair_2qp3 · 2026-01-07

**Summary:**

The paper introduces PromptHash, an instruction-side watermarking framework designed to verify the origin and integrity of prompts given to large language models (LLMs). Unlike output-side watermarking approaches, PromptHash embeds cryptographically bound tags into instructions themselves, aiming to resist adversarial manipulations such as paraphrasing and cross-context splicing. The method combines a position-sensitive hash chain with a semantics-preserving codebook and fuzzy verification, achieving high reported accuracy (TAR >98%, FAR <1%) with negligible latency and token overhead.

**Reviewer Concerns:**

- Codebook Generalizability: The reliance on codebooks raises questions about portability across languages and domains. The paper does not clarify whether codebooks are manually curated, automatically generated, or learned, nor does it test cross-lingual or domain-specific scenarios.
- Evaluation Gaps: While the paper claims the watermark is semantics-preserving, no experiments assess its impact on downstream LLM task performance (e.g., reasoning benchmarks like MMLU or GSM8K). This omission weakens the claim that the watermark does not degrade utility.
- Ambiguity in Core Components: Key elements—particularly the Renderable() function, codebook construction, and fuzzy verification parameters—are underspecified. Reviewers request algorithmic details, pseudo-code, or at least a clear description of how paraphrase tolerance is achieved without compromising security. Without this, reproducibility and interpretability suffer.

**Reviewer Scores:**

Reviewer 3qf4 may have a chance to raise the score.

---

### Decision · Program_Chairs · 2026-01-26

Reject